# Frames of Reference Collectively Organize Space to Influence Attentional Allocation

**DOI:** 10.3390/bs15121713

**Published:** 2025-12-11

**Authors:** Yaohong Liu, Weizhi Nan

**Affiliations:** Department of Psychology and Center for Brain and Cognitive Sciences, School of Education, Guangzhou University, Guangzhou 510006, China; 2112308018@e.gzhu.edu.cn

**Keywords:** spatial cognition, attentional allocation, frame of reference

## Abstract

Spatial cognition refers to how people transform physical spatial information into mental representations and manipulate it to perform further spatial computation and reasoning. Previous research has demonstrated that the frame of reference (FOR) in physical space could distort spatial representations to influence the memory of spatial relations. However, it remains unclear whether FORs could also influence attentional allocation among the spatial representations. To address this issue, we examined the attentional shifting within or between different spatial regions, which were affected by the same versus different FORs. In Experiment 1, a modified double-rectangle cuing paradigm was adopted. Two human figures in complementary colors were presented to establish two object-centered spatial FORs, which divided the external space around the objects into a central region (influenced by two FORs) and two outer-side regions (primarily influenced by a single FOR). Cues and targets were presented in the same region or different regions. Results showed faster attentional shifting within the same region than between different regions. In Experiment 2, one human figure was replaced as a cross, and the within-region advantage was replicated. Overall, these findings suggest that object-centered FORs could be employed to collectively organize space and guide attentional allocation in the external space surrounding objects.

## 1. Introduction

Spatial cognition is a fundamental psychological function that contributes to adaptive interaction with the environment by guiding perception, action planning, environmental evaluation and decision making ([31]). It involves establishing corresponding mental spatial representations based on real-world physical space. However, rather than being a detailed copy of the physical space, these mental spatial representations are constructed by elements in space and their spatial relations relative to a frame of reference (FOR) ([29]; [33]). As a result of this constructive process, the mental space does not always conform to the properties of physical space. Whereas the physical world is three-dimensional, absolute, continuous, and Euclidean, mental space is often fragmented, relative, partial, distorted, and non-Euclidean ([11]; [21]; [23]; [29]; [32]).

Research on object-based attention (OBA) indicates that the structure of objects in physical space plays a critical role in shaping mental spatial representation ([2]; [9]; [26]). As a mechanism of selective attention, OBA refers to the principle that attention is constrained and guided by object structure. Specifically, when attention is directed to one part of an object, the rest of the attended object is also allocated more attention ([5]; [7]; [26]). To account for the modulation of object structure on attentional allocation, [34] ([34]) proposed a grouped array account, which suggests that locations are perceptually grouped or organized by object structure, and that the spread of spatial attention is limited by the contours and boundaries of the attended object ([13]; [26]). This account stresses the important role of the structure of objects on grouping space for the allocation of attention. Various studies have explored different forms of object structure involved in OBA. In the classic double-rectangle cuing paradigm ([9]), participants respond faster to targets at the uncued location within the cued object (invalid within-object) compared to equidistant locations on the uncued object (invalid between-object), demonstrating an object-based facilitation effect. In addition, many other forms of object structure can also trigger object-based attention, such as objects defined by Gestalt principles ([8]), semantically meaningful combinations of words ([19]; [36]), socially interactive objects ([35]), and statistically defined visual chunks ([17]). Collectively, these findings suggest that object structure serves to integrate fragmented visual space, thereby shaping psychological space and influencing the allocation of attention across the visual field.

In addition to object structure, the spatial relations among objects in physical space also influence the formation of psychological space. [12] ([12]) demonstrated that spatial representations are biased and distorted by subjective spatial categories. Specifically, participants overestimated distances between landmarks belonging to different categories while underestimating those within the same category, even in the absence of explicit boundaries ([29]). Similarly, spatial relations based on real-world regularities facilitate inter-object grouping, as in the common perception of a lamp positioned above a table. Such grouping integrates objects into higher-level units, thereby reducing representational competition and improving the efficiency of real-world perception ([14]).

Furthermore, [29] ([29]) investigated whether multiple intrinsic FORs could represent objects’ relations and serve as hierarchical structures to shape mental spatial representation. With roots in psycholinguistic research, the FORs are classified into egocentric (viewer-based), allocentric (environment-based), and intrinsic (object-based) ([4]; [18]). In their research, participants were asked to memorize the spatial relations of a 3 × 3 object array in which the central object varied in salience. Faster responses to relations involving the central object indicated that the intrinsic FORs of central objects distorted spatial representation, and this effect was amplified with higher salience of the central objects. In addition, a follow-up experiment with a 5 × 3 object array containing two salient central objects revealed that each central object established its own intrinsic FOR. However, relations between the two central objects elicited slower responses, indicating competition between intrinsic FORs. These findings suggest that multiple intrinsic FORs can be simultaneously constructed and interact to hierarchically organize and distort spatial representations ([29]).

While [29] ([29]) suggested that object-centered FORs could influence memory for spatial relations, it remains unclear whether object-centered FORs could also play a role in structuring space and guiding attention. Previous studies have shown that attentional allocation in visual space can be influenced by object structure, which integrates the internal space occupied by objects. Differently, object-centered FORs may enable the grouping of objects’ external space through representing their spatial relations and collectively influence the allocation of attention in the external space.

To test this hypothesis, the present study modified the double-rectangle cuing paradigm across two experiments. In Experiment 1, we introduced two human figures in complementary colors to establish two object-centered FORs. More importantly, due to the different impact of these object-centered FORs, the external space was roughly divided into the outer-side region (the outer sides of each object, primarily influenced by a single FOR) and the central region (the space between objects, primarily influenced by two FORs), where cues and targets were positioned. Our hypothesis is that attentional shifting occurring within the same region is significantly faster than that occurring between different regions. Experiment 2 increased the disparity between object representations by replacing one figure with a cross, investigating whether differences in object representations influence the spatial grouping.

## 2. Experiment 1

Figures endowed with stronger social attributes are more likely to gain prioritized cognitive processing and to strengthen sensory representations of objects ([20]; [25]; [27]; [28]). Thus, compared to the simple rectangles in the classic double-rectangle cuing paradigm, Experiment 1 introduced two human figures in complementary colors to establish two object-centered FORs. Additionally, Experiment 1 placed cues and targets in a 3 × 2 array of positions surrounding objects to examine whether attention is differentially allocated across distinct regions of external space. If object-centered FORs integrate the surrounding spatial information into segments and distort the mental spatial representation, we should observe a difference in attentional shifting within same region versus between different regions (participants respond more rapidly in the within-region condition than in the between-region condition).

### 2.1. Methods

#### 2.1.1. Participants

The sample size was determined by a priori power analysis using the G*Power 3.1 program ([10]). We set the alpha level at 0.05 and found that 24 observers would provide 0.80 power to find a medium effect size (0.6) for the paired samples *t*-test.

Experiment 1 recruited 32 undergraduate students (aged 20 ± 1.79 years, 23 females) from Guangzhou University. All participants reported that they had no neurological or psychiatric history and normal or corrected-to-normal vision. They were naive to the experiment’s purpose. Each participant voluntarily enrolled and signed an informed consent form prior to the experiments. This study was approved by the Institutional Review Board of the Educational School, Guangzhou University.

#### 2.1.2. Apparatus and Stimuli

The participants were seated in a silent room approximately 60 cm away from a Dell monitor (screen size = 60 × 35.5 cm; resolution = 1024 × 768 pixels; refresh rate = 60 Hz). Two figures were subtended 10.11° × 8.81° (vertical condition) and 8.81° × 10.11° (horizontal condition). The cue (0.76° × 0.76°) was a black star (✱) and the target (0.55° × 0.55°) was a black square (■). Fixation (0.29° × 0.29°) was a plus sign presenting in the center of the screen. The distance between cues and targets was 14.04°. And the distances from cues and targets to the fixation was 6.65° (locations: 3/4) and 15.38° (locations: 1/2/5/6) (see Figure 1A). All stimuli were presented on a gray (RGB: 128, 128, 128) background. Stimulus presentation and manual response measurements were controlled by E-Prime 2.0 software (Psychological Software Tools, Inc., Pittsburgh, PA, USA).

#### 2.1.3. Procedure and Design

Experiment 1 was a single factor (cue validity: valid, within-region, between-region) within-subject design. Cues and targets appeared in six locations around the objects (see Figure 1B). We set three types of cue validity: valid (cues and targets appeared in the same location, e.g., cues and targets both occurred in position 1), within-region (cues and targets appeared in different locations while within the same outer-side region or central region, e.g., cues occurred in position 1 while targets occurred in position 2, or cues occurred in position 3 while targets occurred in position 4), between-region (cues and targets occurred in locations between different regions, e.g., cues occurred in position 1 while targets occurred in position 3). The balance condition (e.g., cues occurred in position 1 while targets occurred in the position 4/5/6) was set to balance the cue validity. The cue-to-target distance was equal in the within-region and between-region conditions. The orientation of objects (vertical vs. horizontal) was counterbalanced as a within-subject factor (see Figure 1C) as previous research has demonstrated that attentional shifts are faster along the horizontal than the vertical meridian, known as the meridian effect ([1]; [3]; [24]).

Experiment 1 consisted of 384 trials overall, including 272 target-present trials and 112 target-absent trials (catch trials). For target-present trials, there were 144 trials for valid condition, 48 trials for each of within-region and between-region conditions, and 32 trials for balance condition.

Each trial began with a 500 ms fixation, followed by two human figures for 1000 ms. Then a cue appeared in any of six positions randomly for 100 ms. After a 100 ms inter-stimulus interval (ISI), the target (or no-target in the catch trials) appeared for 2000 ms or until the participant responses (see Figure 1A). Participants were instructed to stare at the centered location of fixation even when the fixation disappeared throughout the whole trial, and to press the “SPACE” button as rapidly and accurately as possible when the target appeared. The entire experiment comprised 16 blocks (vertical: 8 blocks, horizontal: 8 blocks), each containing 24 trials, and there was a rest period between each block. Before the formal test, participants completed a practice session that required them to make at least 20 correct responses in a row. The whole experiment lasted approximately 30 min.

#### 2.1.4. Statistical Analysis

The response times (RTs) for horizontal and vertical conditions were combined to compare the difference in the within-region and the between-region conditions.

All participants were included in the analysis. The catch trials and balance trials were removed. The mean accurate rate for catch trials was 92.7%. Trials with RTs faster than 150 ms or slower than 1500 ms, and beyond three standard deviations in each condition (1.8%) were discarded. To compare the attentional shifting within same region and between different regions, a paired samples *t*-test was conducted on RTs with within-region and between-region conditions. To examine the spatial-based attention, within-region and between-region conditions were coalesced as the invalid condition, and a paired samples *t*-test was conducted on the RTs with valid and invalid conditions.

### 2.2. Results

The difference between the within-region and between-region conditions was significant, *t*(31) = 2.66, *p* = 0.012, Cohen’s *d* = 0.47, 95% CI [0.10, 0.83], with shorter RTs in the within-FOR condition (420 ± 12 ms) relative to those in the between-FOR condition (430 ± 13 ms), see Figure 2A.

The difference between valid and invalid conditions was significant, *t*(31) = 4.46, *p* < 0.001, Cohen’s *d* = 0.79, 95% CI [0.39, 1.18], with shorter RTs in the valid condition (405 ± 11 ms) relative to those in the invalid condition (425 ± 12 ms), see Figure 2A.

### 2.3. Discussion

The results of Experiment 1 were consistent with our expectations, in that we not only observed the classic spatial-based attention (i.e., shorter RTs in valid conditions compared to invalid conditions) but also observed shorter RTs in the within-region condition compared to the between-region condition. These findings aligned with the suggestions proposed by [29] ([29]) that the external space is segmented and divided into distinct spatial regions under the influence of object-centered FORs. Consequently, this spatial organization may influence attentional allocation in the visual field. Specifically, attentional shifting within the same region was more efficient than that between different regions.

## 3. Experiment 2

Building on Experiment 1, Experiment 2 was designed to examine the robustness of the observed within-region advantage by introducing a disparity of object representations. To this end, Experiment 2 replaced one of the human figure stimuli with a cross-shaped figure to increase the disparity of object representations. We expected that the within-region advantage would remain robust under this manipulation.

### 3.1. Methods

Experiment 2 recruited 27 undergraduate students (aged 21 ± 1.95 years, 19 females) from Guangzhou University. The same power calculation method as for Experiment 1 was used to determine the sample size for Experiment 2. The stimuli and procedure for Experiment 2 were the same as those of Experiment 1, except for replacing one of the human figure stimuli with a cross-shaped figure of equal total area, which was composed of two rectangles (see Figure 1C).

The method of statistical analysis used for Experiment 2 was the same as that in Experiment 1. All participants were included in the analysis. The catch trials and balance trials were removed. The mean accurate rate for catch trials was 94.3%. Trials with RTs faster than 150 ms or slower than 1500 ms, and beyond three standard deviations in each condition (2.0%) were discarded.

### 3.2. Results

The difference between the within-region and between-region conditions was significant, *t*(26) = 2.51, *p* = 0.018, Cohen’s *d* = 0.49, 95% CI [0.08, 0.88], with shorter RTs in the within-FOR condition (413 ± 9 ms) relative to that in the between-FOR condition (421 ± 10 ms), see Figure 2B.

The difference between the valid and invalid conditions was significant, *t*(26) = 4.62, *p* < 0.001, Cohen’s *d* = 0.89, 95% CI [0.44, 1.33], with shorter RTs in the valid condition (394 ± 7 ms) relative to those in the invalid condition (417 ± 10 ms), see Figure 2B.

### 3.3. Discussion

The results of Experiment 2 showed that after increasing the disparity of object representations, a significant difference between the within-region and between-region conditions was still observed. These results replicated the findings of Experiment 1 and supported the notion that object-centered FORs collectively influence spatial organization, subsequently shaping attentional allocation in objects’ external space.

## 4. General Discussion

The present study investigated how object-centered FORs distort spatial representation and consequently influence the allocation of spatial attention. Across two experiments, we found that the external space surrounding objects was structured by object-centered FORs into two distinct regions: outer-side region (the outer sides of each object) and central region (the space between objects). These findings are consistent with the “Frame of Reference-based Maps of Salience (FORMS)” theory, which states that multiple representations with distinctive FORs integrate spatial information into each respective subset to construct the psychological space ([22]; [29]; [30]). Accordingly, humans represent spatial information simultaneously through multiple FORs, and human performance is affected by the interaction among these coexisting FOR-based representations ([6]; [22]). Consistent with this view, the present study demonstrates that object-centered FORs collectively integrate surrounding spatial information, thereby constructing mental spatial representations.

Previous research has also examined the attentional allocation in the external space of objects. [15] ([15]) presented participants with a single object and, after the object was cued, a target appeared in the external space. The results found that response times were faster when the target was closer to the object’s centroid, suggesting the presence of an attentional gradient centered on the object’s center of mass in the external space. This finding was interpreted as that attention spreads not only within the cued object but also into the external space through overlapping receptive fields and the top down influence of higher-level object representation, forming an attentional gradient beyond object boundaries ([13]; [16]). In contrast, the present study emphasizes the role of object-centered FORs in integrating the external space and guiding attentional allocation. Critically, the manipulation of positioning both cues and targets in the external space did not directly induce object-based attention, suggesting that beyond cross-boundary attentional spreading effect driven by objects, the external space could be integrated and segmented by object-centered FORs, without the activation of objects.

In a subsequent study, [16] ([16]) introduced the similarity of object feature to examine interactions among object-based attention (OBA), spatial-based attention (SBA), and feature-based attention (FBA). They found that when objects shared highly similar features, the activation of the cued object would spread to the uncued object, conferring a comparable attentional advantage. However, this effect disappeared when the similarity of object features was low. These findings suggest that the interaction among OBA, SBA, and FBA promotes a more holistic organization of visual scenes ([16]). In contrast, Experiment 2 showed a robust within-region advantage under reduced feature similarity. One possible reason is that the absence of OBA in our paradigm prevented FBA from interacting with OBA, making it unlikely for the feature similarity to modulate attentional allocation.

## 5. Conclusions

Taken together, this study investigated the crucial role of object-centered FORs in modulating spatial integration and segmentation and shaping spatial attentional allocation. These results not only provide evidence for the FORMS theory, but also reveal a broader, scene-wide organization of attentional allocation. This contributes to a deeper understanding of how attention is dynamically and flexibly organized in complex environments, enabling humans to efficiently distribute limited cognitive resources amid perceptual complexity.

## Figures and Tables

**Figure 1 behavsci-15-01713-f001:**
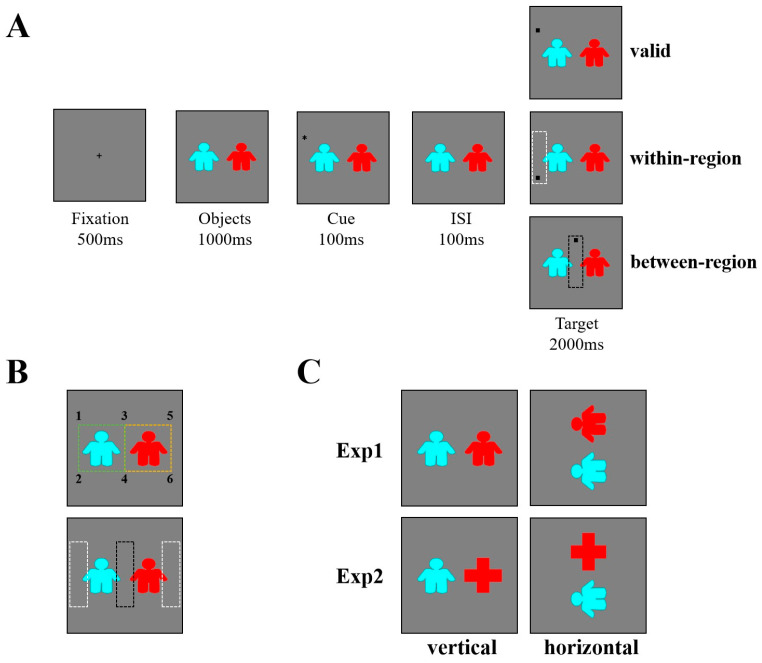
(**A**) Trial sequences for Experiments 1 and 2. (**B**) Upper panel: numbers 1–6 indicate the positions of cues and targets. Valid condition: cue–target pairs were 1-1, 2-2, 3-3, 4-4, 5-5, 6-6. Within-region condition: cue–target pairs were 1-2, 2-1, 3-4, 4-3, 5-6, 6-5. Between-region condition: cue–target pairs were 1-3, 2-4, 3-1, 3-5, 4-2, 4-6, 5-3, 6-4. The green and yellow dotted squares indicate the same shifting distance for within-region and between-region conditions. Numbers and squares were not displayed during experiments; Lower panel: the white dotted boxes indicate the outer-side region, and the black dotted box indicates the central region. All boxes were not displayed during the experiment. (**C**) Stimuli of objects: colored human figures in Experiment 1; a colored human figure and a cross in Experiment 2. Each stimulus includes both vertical and horizontal conditions.

**Figure 2 behavsci-15-01713-f002:**
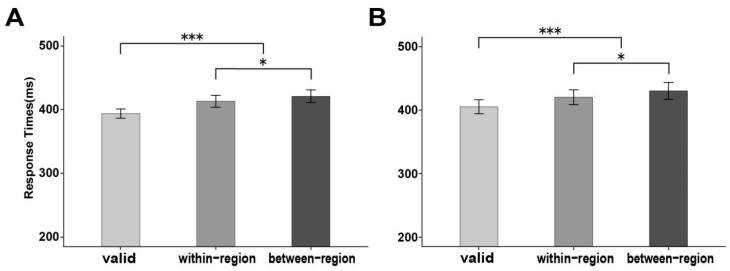
Error bars represent the standard errors of the mean (SE). *** *p* < 0.001, * *p* < 0.05. (**A**) RTs for three conditions of cue validity in Experiment 1. (**B**) RTs for three conditions of cue validity in Experiment 2.

## Data Availability

Data from this study are available at https://osf.io/r2w53/ (accessed on 8 December 2025).

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
