# Peer review of "Frames of Reference Collectively Organize Space to Influence Attentional Allocation"

_behavsci, 2025, doi:10.3390/bs15121713_

Round 1

Reviewer 1 Report

Comments and Suggestions for Authors

This is on the whole an interesting attempt to address an important question.  My suspicion is that the authors’ hypothesis is correct, but I have one point of serious concern about the experimental design (point b below), one more mechanical point about the description of the experiments (point a below), and one more informational query (point c below).

a) There is an inconsistency between the verbal description of the locations for cues and targets (which specifies 8 slots) and the diagrammatic form in figure 1 (which shows 6 slots).

b) The discussion section (lines 247-51) notes that researchers have found “attentional gradients” in related contexts.  This suggests the possibility of an alternative interpretation of the findings of the present paper, which had already occurred to me in any case.  The authors argue that their findings are the result of schematic segmentation of space.  But it seems to me that they are also in principle compatible with a the effects of a continuous gradient of distance between cue and target without regard to structured frames of reference.  The inconsistency mentioned above makes it hard to discuss precisely how to address this, but for the sake of argument, let’s assume that the experiment was actually done with the 6-position system illustrated in figure 1.  Do the authors have enough data to break out “between regions” instances which are 1 vs. 2 regions away?  And are there any effects in either the between- or the within-regions cases if the cue and the target are not on the same horizontal line?

c) One much less important point: the experiments both use both a “vertical” and a “horizontal” condition, but the analysis does not seem to address why this was done or whether they showed distinctive results.  Could the authors say more about this?

Reviewer 2 Report

Comments and Suggestions for Authors

This paper reports two relatively simple experiments aimed at confirming that objects present in the visual field tend to generate frames of reference that facilitate the detection of events that occur within them after a cue has moved covert attention to that same frame of reference. The results confirm that if covert attention is called, by the presentation of a cue, in a region of the visual field where a relevant object is present, the appearance of another event in the same area is perceived more promptly, but the presence of another object in another region of the visual field interferes with this phenomenon .

Essentially, this suggests that relevant objects in the visual field interfere among them in calling for attention; this had been clearly demonstrated by previous works – cited in the paper – that showed that "frames of reference in the physical space could distort the spatial representations to influence memory of spatial relations". Thus, elaborating on the cognitive relevance of these results appears to this reviewer as forcing the interpretation of the data well beyond what they actually tell.

In summary, the reported experiments are correctly designed and analysed. The results are not particularly relevant and the strong conclusions proposed by the Authors are more based on conceptions that derive from the literature than on the data themselves.

Specific observations:
legend of fig. 1 must be rewritten. It refers to A, B, C which are not present in the figure. Also, please specify "invalid within-region" and "invalid between-region"
line 170: 3 sd before or after eliminating RT<150 ms or >1500 ms?
line 193: this is somewhat presumptuous, as if these data are sufficient to "demonstrate" that external space is segmented et cetera, then this had already been demonstrated; these data simply corroborate previous suggestions
line 213: see above for line 170
section 3.2 Results: where there any difference between invalid trials in the region of the human figure or the one of the cross? the whole point of experiment 2 (w.r.t. experiment 1) seems to be that the two stimuli are different...
lines 278-9: it does not seem to me that this claim is either tenable or clearly justified and argued for 

Reviewer 3 Report

Comments and Suggestions for Authors

I thank the editors for the trust, and the authors for their interesting work as reflected in their paper. I believe it is a valuable work, a worth publishing paper, nevertheless, if you excuse me, I will allow myself to make some suggestions regarding some concerns in order to make the paper more clear to me, in the first place and, in my understanding, to the academic community:

  1. FORMS theory cited in line 278 of page 7 I guess is an acronym, not revealed, of the "Frame of Reference based Maps of Salience" theory referenced in page 6, line 235. It would be worthwhile to link the two of them.
  2. Subsections in figure 1 are not signaled: where is A, B, C, etc. inside the figure.
  3. I had a hard time to figure out what was expected from the volunteers to do in the experiments, in the figure and in the text, I strongly encourage the authors to explicitly state what is the intended response the participants should do and what is expected from each of the theories to be happening ("volunteers should respond more rapidly to the X than to the Y"). That would make much more easier the task for the readers to understand.

Reviewer 4 Report

Comments and Suggestions for Authors

In this set of experiments the authors investigate how object-centered frame of reference modulates spatial attention using attentional cuing with human figures. They found that on trials with a non-valid cue, participants were faster to respond to targets that were within-region to the cue than between-region. This demonstrated that external space is segmented and divided into spatial regions and is influenced by object-centered frames of reference, contributing to our broader understanding of how attention is dynamically organized in complex environments.

Overall the article was clearly and concisely written. The authors aptly summarized current theories around object based attention and psychological space. The experiments were designed to examine whether attention can shift faster when it occurs in the same region as the frame of reference object than when attention is shifted into a region between two frames of references of objects (where there are competing frames of reference). The experiments were well-designed and the conclusions appropriate. This was a neat little set of experiments that help to advance our understanding of how object based attention works. 

I recommend that the article be accepted in present form.

Round 2

Reviewer 1 Report

Comments and Suggestions for Authors

The revised paper adequately responds to my concerns with the original draft.